# Role of Eddy Diffusion in the Delayed Ionospheric Response to Solar Flux Changes

Rajesh Vaishnav[1], Christoph Jacobi[1], Jens Berdermann[2], Mihail Codrescu[3], and Erik Schmölter[2]

[1]Leipzig Institute for Meteorology, Universität Leipzig, Stephanstr. 3, 04103 Leipzig, Germany
[2]German Aerospace Center, Kalkhorstweg 53, 17235 Neustrelitz, Germany
[3]Space Weather Prediction Centre, National Oceanic and Atmospheric Administration, Boulder, Colorado, USA

**Correspondence:** Rajesh Vaishnav (rajesh_ishwardas.vaishnav@uni-leipzig.de)

**Abstract.** Simulations of the ionospheric response to solar flux changes driven by the twenty-seven days solar rotation have been performed using the global 3-D Coupled Thermosphere/Ionosphere Plasmasphere electrodynamics (CTIPe) physics-based numerical model. Using the F10.7 index as a proxy for solar EUV variations in the model, the ionospheric delay at the solar rotation period is well reproduced and amounts to about 1 day, which is consistent with satellite and in-situ measurements. From mechanistic CTIPe studies with reduced and increased eddy diffusion, we conclude that the eddy diffusion is an important factor that influences the delay of the ionospheric total electron content (TEC). We observed that the peak response time of the atomic oxygen to molecular nitrogen ratio to the solar EUV flux changes quickly during the increased eddy diffusion compared with weaker eddy diffusion. These results suggest that an increase in the eddy diffusion leads to faster transport processes and an increased loss rates resulting in a decrease in the ionospheric time delay. Furthermore, we found that an increase in solar activity leads to an enhanced ionospheric delay. At low latitudes, the influence of solar activity is stronger because EUV radiation drives ionization processes that lead to compositional changes. Therefore, the combined effect of eddy diffusion and solar activity leads to a longer delay in the low and mid latitude region.

## 1 Introduction

The solar activity plays a significant role in controlling the variations in the thermosphere/ionosphere (T/I) system, in particular through solar Extreme Ultraviolet (EUV) and Ultraviolet (UV) radiation and their variability. In addition, there are several factors which control the behaviour of the T/I system, such as tidal and gravity wave forcing from the lower atmosphere, neutral winds, and related currents in the ionosphere. These are especially predominant during low solar activity, leading to reduced correlation of solar flux and ionospheric electron density then (Vaishnav et al., 2019). The ionosphere itself is created through photoionization of the major constituents (atomic oxygen, molecular nitrogen, and molecular oxygen), while photodissociation may change the mixing ratios of these constituents (especially atomic oxygen), leading to modifications of the ionisation rates.

Due to these varying ionization rates for different atoms and molecules, a series of layers of electron density forms, known as D, E, F1, and F2 regions. The maximum peak of electron density is observed in the F2 region. The F2 region electron densities mainly depend on photo-chemical processes such as photodissociation, photoionization, and loss by recombination with molecular nitrogen, and transport processes such as neutral wind and diffusion. On top of this, there are many processes

which can drive or disturb the ionospheric ion distribution, such as diffusion, transport, cooling, and heating mechanisms. Transport can be divided into three main categories, namely eddy diffusion, molecular diffusion, and advection processes (Brasseur and Solomon, 2006). The F2 region is strongly influenced by the global thermospheric circulation (Rishbeth, 1998).

    The physical mechanism of the delayed ionospheric response that cannot be explained with solar variations, seasonal variations or changes due to geomagnetic activity. We also cannot explain the delay with photoionization and photo-dissociation

processes alone. This has been discussed in several studies (Jakowski et al., 1991; Schmölter et al., 2018, 2020, & references therein) and the importance of the T/I coupling was pointed out there. The T/I coupling is important for the delay. This has been mentioned in several studies by now but was scarcely investigated. The most important impacts we would expect due to this coupling are compositional changes that can impact the major processes, and this could be due to gravity waves induced diffusion, etc.

The lower thermospheric composition is not only influenced by gravity waves, but also by other parts of the lower atmospheric wave spectrum including atmospheric tides and planetary waves. The main source of eddy diffusion is breaking of gravity waves. Gravity waves are usually generated in the lower atmosphere by various mechanisms such as convection, wind shears, storms, and airflow over mountains. Their amplification and wave breaking due to instabilities produces mixing (Li et al., 2005). Atmospheric tidal and planetary wave activity can also significantly contribute to eddy diffusion. For example,

the tides induce a net transport of atomic oxygen via the mean meridional circulation generated by tidal dissipation (Jones et al., 2014a, b). Above the mesopause, the intradiurnal variability associated with atmospheric tides strongly affects the transport of NOx (Meraner and Schmidt, 2016), and seasonally varying gravity wave and tidal mixing influence in the mesosphere-lower thermosphere (MLT) region (Qian et al., 2013). Moreover, the possible role in the semiannual oscillation in thermospheric mass density is discussed by Jones et al. (2018). Siskind et al. (2014) showed that the vertical transport by nonmigrating tides

causes a significant reduction in the calculated peak electron density of the ionospheric F2 layer.

    Several studies have reported the influence of gravity waves and turbulence on the T/I composition and calculated the eddy diffusion coefficient in the MLT region (Kirchhoff and Clemesha, 1983; Sasi and Vijayan, 2001; Swenson et al., 2019). Based on radar measurements, Kirchhoff and Clemesha (1983) calculated a minimum (maximum) eddy diffusion coefficient of 45 (123) $m^2 s^{-1}$ during fall (summer). Similarly, Sasi and Vijayan (2001) used Doppler radar observations and show that the eddy

diffusion varies from 25 to 300 $m^2 s^{-1}$ during September and June.

    Turbulent mixing is an important process affecting the composition of the T/I system. The effect of turbulence on different minor and major species has been discussed on several occasions (e.g., Keneshea and Zimmerman, 1970; Shimazaki, 1971; Chandra and Sinha, 1974; Rishbeth et al., 1987; Rees and Fuller-Rowell, 1988; Fuller-Rowell and Rees, 1992; Danilov and Konstantinova, 2014; Pilinski and Crowley, 2015; Swenson et al., 2018). Various coupled models have been developed to

understand the T/I region variations considering the availability of experimental and theoretical knowledge. Earlier 1-D models, which include eddy diffusion coefficients have been used to model the T/I region (e.g., Colegrove et al., 1965; Shimazaki, 1971; Jakowski et al., 1991). Nowadays, more improved, 3-D models like the Coupled Thermosphere Ionosphere Plasmasphere Electrodynamics (CTIPe) model (Fuller-Rowell and Rees, 1980) or the National Center for Atmospheric Research (NCAR) Thermosphere-Ionosphere-Electrodynamics General Circulation Model (TIE-GCM) (Richmond et al., 1992) are available to

explore the dynamics of the T/I region. These models cannot be expected to exactly reproduce the real ionospheric variability due to limited knowledge of various processes in the T/I region and their corresponding inputs (e.g., Shim et al., 2011; Codrescu et al., 2012), but they are capable of providing insight into relevant dynamical processes in the T/I.

Rees and Fuller-Rowell (1988) used a sinusoidal eddy turbulence profile and analysed the effect of eddy turbulence on temperature, atomic oxygen, and nitric oxide. They showed that an increase in turbulence near the mesopause leads to an increase in atomic oxygen and nitric oxide. This leads to a change in the thermal structure by strongly modifying the gravity wave flux.

The solar radiation reaching the Earth exhibits a periodicity of about 27 days owing to the solar rotation. As a result, the T/I system also varies with this periodicity. Many studies revealed a delay in ionospheric parameters such as total electron content (TEC, given in TEC units, 1 TECU=$10^{16}$ electrons $m^{-2}$), electron density, peak electron density of $F_2$ region (NmF2, $cm^{-3}$) and the corresponding height (hmF2, km), to the 27 day solar flux variation (Jakowski et al., 1991; Liu et al., 2006; Afraimovich et al., 2008; Lee et al., 2012; Anderson and Hawkins, 2016; Jacobi et al., 2016; Schmölter et al., 2018, 2020, 2021; Vaishnav et al., 2018, 2021; Ren et al., 2018, & references therein). Most of the studies found an ionospheric delay of about 1-2 d, with a possible uncertainty of about half a day. Schmölter et al. (2018), using high temporal resolution data, found an ionospheric delay of about 17-19 h using TEC and Geostationary Operational Environmental Satellites (GOES) EUV datasets. The detailed seasonal and spatial effect on the ionospheric delay was studied by Schmölter et al. (2020). Their study revealed a strong geomagnetic effects on the ionospheric delay. They also noticed that the delay over Southern Hemispheric stations is larger than over Northern Hemisphere stations.

Numerical simulations using a 1-D model have revealed that the delay might be due to the slow diffusion of atomic oxygen at 180 km height, generated by solar UV radiation in the Schumann-Runge continuum, causing photodissociation of molecular oxygen above the turbopause (Jakowski et al., 1991).

Ren et al. (2018) investigated the ionospheric time delay using observations and simulations with the TIE-GCM model. They disccused the possible role of ion production and loss mechanisms and the $O/N_2$ ratio on the ionospheric delay against the solar EUV flux. A strong effect of geomagnetic activity was reported. The ionospheric response time is controlled by photo-chemical, dynamical, and electrodynamical processes. Ren et al. (2019) suggested that the time delay in thermospheric temperature is due to the difference between the total heating and cooling rates. The study also found a possible role of the general circulations in the upper atmosphere on the time delay. Similarly, the peak response time of the neutral mass density corresponds to the time of equilibrium between the effect of the barometric process and the change in its abundance (Ren et al., 2020). Moreover, Ren et al. (2021) suggest the possible role of geomagnetic activity in the time delay of the thermospheric mass density, which varies with altitude, latitude, and local time.

Despite such a general understanding, however, the exact mechanism of the ionospheric delay needs further investigation. Therefore, here we attempt to quantify the process, which is probably responsible for the ionospheric delay by using the CTIPe model (Fuller-Rowell and Rees, 1980). Vaishnav et al. (2018) indicated that transport processes might play an important role in the ionospheric delay observed in TEC using CTIPe model simulations. Based on this assumption, numerical simulations have been performed to consolidate the preliminary results of Vaishnav et al. (2018) and the hypothesis of Jakowski et al.

(1991), and to explain the physical mechanisms of the ionospheric delay. To understand the role of T/I coupling in this study, we perform model runs changing the eddy diffusion.

An ionospheric delayed response has been investigated by Schmölter et al. (2020) over European stations. They reported an ionospheric delay of about 18 h over these stations. Therefore, in this paper, we emphasis to reproduce and investigate the ionospheric delay response over an European location ($40°N$).

## 2   CTIPe Model Simulations

The CTIPe model is used to understand the influence of eddy diffusion in the neutral composition and its role in the delay mechanism. The CTIPe model is an advanced version of the CTIM model (Fuller-Rowell et al., 1987) and is a global, first principle, non-linear, time-dependent, 3-D, numerical, physics-based coupled thermosphere-ionosphere-plasmasphere model consisting of four fully coupled distinct components, namely, (a) a neutral thermosphere model (Fuller-Rowell and Rees, 1980),
(b) a high-latitude ionosphere convection model (Quegan et al., 1982), (c) a mid- and low-latitude ionosphere plasmasphere model (Millward et al., 1996), and (d) an electrodynamics model (Richmond et al., 1992). The thermosphere component of the CTIPe model solves the continuity, momentum, and energy equations to calculate the wind components, global temperature, and composition.

The transport terms particularly specify the $E \times B$ drift and include ion-neutral interactions under the effect of the mag-
netospheric electric field. The geographic latitude/longitude resolution is 2°/18°. In the vertical direction, the atmosphere is divided into 15 logarithmic pressure levels at an interval of one scale height, starting with a lower boundary at 1 Pa (about 80 km altitude) to above 500 km altitude at pressure level 15. The high-latitude ionosphere (poleward of geomagnetic coordinates 55° N/S) and the mid- and low-latitude ionosphere and plasmasphere are implemented as separate components, and there is an artificial boundary between these two model components. The equations for the neutral thermosphere model are solved
self-consistently with a high-latitude ionosphere model (Quegan et al., 1982). The numerical solution of the composition equation describes transport, turbulence, and diffusion of atomic oxygen, molecular oxygen, and nitrogen (Fuller-Rowell and Rees, 1983). External inputs are needed to run the model, such as solar UV and EUV, Weimer electric field, TIROS/NOAA auroral precipitation (note, however, that particle precipitation is turned off during our simulations), and tidal forcing from the Whole Atmosphere Model (WAM). The F10.7 index (Tapping, 1987) is used as a solar proxy for calculating ionization, heating, and
oxygen dissociation processes. Within CTIPe, a reference solar spectrum based on the EUVAC model (Richards et al., 1994) and the Woods and Rottman (2002) model, driven by variations of input F10.7 is used. The EUVAC model is used for the wavelength range from 5 to 105 nm, and the Woods and Rottman (2002) model from 105 nm to 175 nm. Solar flux is obtained from the reference spectra using the following equation:

$$f(\lambda) = f_{ref}(\lambda)[1 + A(\lambda)(P - 80)] \tag{1}$$

where $f_{ref}$ and A are the reference spectrum and a solar variability factor, and $P = 0.5 \times (F10.7 + F10.7A)$, where F10.7A is the average of F10.7 over 41 days. Detailed information on the CTIPe model is available in Codrescu et al. (2008, 2012).

In this paper, our primary goal is to understand the influence of eddy diffusion on the ionospheric response during the 27 day solar rotation. Therefore, several model runs were performed in this study with different diffusion conditions under different artificial solar activity conditions. Three runs were performed with sinusoidally varying solar activity from 75-125 sfu, keeping all other conditions constant. Constant atmospheric and astronomical conditions of 15 March 2013 were used to perform these experiments.

Several authors have suggested that the eddy diffusion is strongly varying based on the months or seasons (e.g., Kirchhoff and Clemesha, 1983; Sasi and Vijayan, 2001; Swenson et al., 2019). Therefore, the experiments were performed by using an eddy diffusion coefficient $K_T$, which amounts to 75%, 100%, and 125% of the original values in the model, and we refer to these runs as $K_T \times 0.75$, $K_T \times 1.0$, and $K_T \times 1.25$, where $K_T \times 1.0$ represents the reference run.

## 3   Mechanistic Studies

In the CTIPe model, the T/I composition is calculated by combining the continuity equation with the diffusion equation. The model estimates changes in the composition of the major species ($O$, $O_2$, and $N_2$) self-consistently, including wind and temperature (Fuller-Rowell and Rees, 1983), as well as molecular diffusion, production, and loss mechanisms.

The continuity equation for the mass mixing ratio, $\psi_i = (n_i \cdot m_i)/\rho$ of the i-th species, with $n_i$ as number density, $m_i$ as the molecular mass, and $\rho$ as atmospheric density, may be written as:

$$\frac{\partial \psi_i}{\partial t} = \frac{1}{\rho}(m_i S_i) - \boldsymbol{V} \cdot \nabla \psi_i - \omega \frac{\partial \psi_i}{\partial p} - \frac{1}{\rho}\nabla \cdot (n_i m_i \boldsymbol{C_i}) + \frac{1}{\rho}\nabla \cdot (K_T n \nabla m \psi_i), \tag{2}$$

where $S_i$ represents sources and sinks of the species, $K_T$ is the eddy diffusion coefficient, $\boldsymbol{V}$ is the horizontal neutral wind vector, $n$ is the total number density, $m$ is the mean molecular mass and $\boldsymbol{C_i}$ is the diffusion velocity of the i-th species. The terms on the right-hand side of equation (2) are, in their respective order, sources and sinks of species, horizontal advection, vertical advection, molecular diffusion, and eddy diffusion.

The mathematical form of the eddy diffusion coefficient $K_T$ used in the CTIPe model as a function of height is given by Shimazaki (1971) and Fuller-Rowell and Rees (1992):

$$K_T = D \exp(-A_1(h - h_o)^2) \quad h \geq h_o, \tag{3}$$

$$K_T = (D - D_o)\exp(-A_2(h - h_o)^2) + D_o \exp(-A_3(h - h_o)) \quad h \leq h_o. \tag{4}$$

A peak value of $D = 150\ m^2 s^{-1}$ at $h_o = 105$ km altitude and $D_o = 100\ m^2 s^{-1}$ is used for the $K_T \times 1.0$ reference run. The shape parameters $A_1 = 0.03$, $A_2 = 0.03$, and $A_3 = 0.05$ are taken from Shimazaki (1971). As pointed out by Fuller-Rowell and Rees (1992), eddy diffusion has the greatest influence on atomic oxygen and nitric oxide in the lower thermosphere. A detailed description of the chemistry of the major species is available in Fuller-Rowell (1984).

In our experiments, the CTIPe model was first run with constant F10.7 input for ten days to achieve a diurnally reproducible condition, and after this spin-up, F10.7 was modified for 27 days using a sine function:

$$F10.7(t) = 100 - 25\cos\left(\frac{2\pi t}{27}\right),$$

(5)

where t represents the time in days.

The various terms of the composition equation are shown in Figure 1 for the noontime (12 UT) for the atomic oxygen mass mixing ratio ($\psi_O$) at $40°N/18°E$ in the reference run. The Figure shows the behavior of all terms at pressure level 12 (260 km). The vertical red dashed line shows the maximum of the input solar flux as per equation 2. Figure 1(a) shows that the molecular diffusion term shows a delay of less than one day for $\psi_O$. The horizontal and vertical advection are decreasing with the increasing input solar flux with a delay of less than one day and one day, respectively (Figure 1(b)). Similar variations can be seen in the chemical production and loss terms (source and sink term). The delay between production and loss is about 1-2 days in the case of $\psi_O$, as shown in Figure 1(c). The change in the production term in the composition equation is based on the photoionization processes contributing to $\psi_O$.

Ren et al. (2020) discussed the physics behind the time delay in different thermospheric neutrals. They found that the peak response time of the mass density of the neutrals ($O$ and $N_2$) corresponds to the time of equilibrium between the effect of the barometric process and the change in their abundance.

Figure 2 shows the daily zonal mean TEC for three different runs with different eddy diffusion coefficients as a function of time and magnetic latitude. The zonal averages represent the average TEC values over all longitudes at a specific magnetic latitude. In the CTIPe model, TEC is calculated over the altitude range from 80 km to 2000 km. In Figure 2, the zonal mean TEC is shown by the contours, and the white curves show the corresponding variability of the F10.7 index. Here, moderate solar activity conditions (75-125 sfu) have been used.

The daily zonal mean TEC show the overall effect of solar flux on the T/I system, since we used constant atmospheric and astronomical conditions for these simulations.

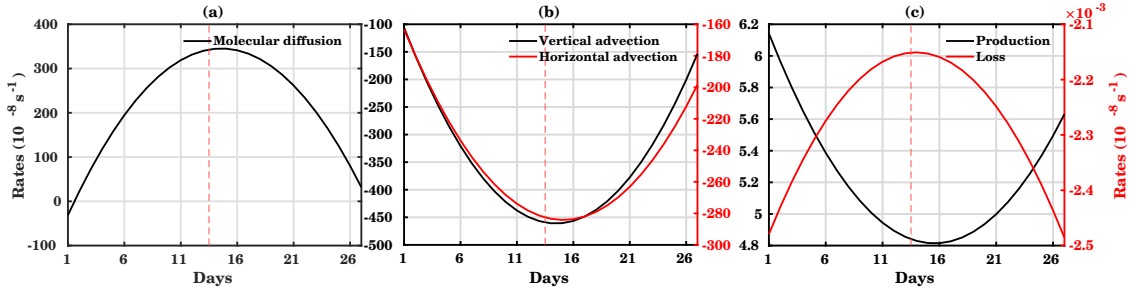

**Figure 1.** Time series of different terms: (a) molecular diffusion, (b) horizontal advection and vertical advection, and (c) production and loss for atomic oxygen mass mixing ratio. Both the y-axis marked with the corresponding color. The vertical red dashed line represents the middle of the $13^{th}$ model day. All the parameters are plotted for pressure level 12 (260 km) for noontime at $40°N/18°E$.

The results from the reference run $K_T \times 1.0$, with the original value of the eddy diffusion coefficient, are shown in Figure 2(b). The simulations reproduce the real latitudinal as well as temporal variations with the variability in the solar flux. The zonal mean TEC distributions are symmetric around the equator, with maximum amplitudes of about 70 TECU. The TEC values decrease towards the high latitudes. The distribution of TEC highly depends on the ionization of neutrals and various processes such as transport and recombination. The TEC amplitude variations reflect the effects of solar activity and compositional changes.

The $K_T \times 0.75$ run results are shown in Figure 2(a). It shows an increase of TEC in the low- to mid-latitude region in comparison to the reference run. The reduction of turbulence leads to slower transport and an increase in TEC. Figure 2(c) shows the zonal mean TEC for the $K_T \times 1.25$ run. In comparison to the reference run, TEC is reduced by a significant amount. These results show that eddy diffusion has a direct impact on TEC.

Figure 3(a) shows the global mean TEC (GTEC) as simulated by the three different runs along with the F10.7 index (white curve). It shows an obvious 27 day variation of GTEC corresponding to the F10.7 index variations, but with a slightly different delay for the different runs. The GTEC values vary from about 8 TECU to maximum values of about 15 TECU for the reference run corresponding to the solar flux variation. It can be seen that TEC increases linearly with F10.7. In comparison to the reference run, TEC values decreased significantly for the increased eddy diffusion condition, while it is increased for the reduced eddy diffusion conditions (see also Figure 2).

The model input F10.7 index has been calculated according to Eq.(5), but as hourly values in order to calculate the delay and cross-correlation between GTEC and F10.7, which are shown in Figure 3(b).

For the reference run $K_T \times 1.0$, the delay is about 24 h, which is close to the value derived from observations as reported by Schmölter et al. (2018, 2020). Therefore, the model is capable of reproducing the observed ionospheric delay. In the case of reduced eddy diffusion to 75% of the original value in run $K_T \times 0.75$, the delay is somewhat longer (about 25 h). This indicates that the delay increases due to the slower transport processes in this run. In line with this, with increased transport in the run $K_T \times 1.25$, the delay reduces to 20 h. These results suggest that an increase in the eddy diffusion leads to faster transport processes and an increased loss rate, resulting in a decrease of the ionospheric time delay. The loss rates are discussed

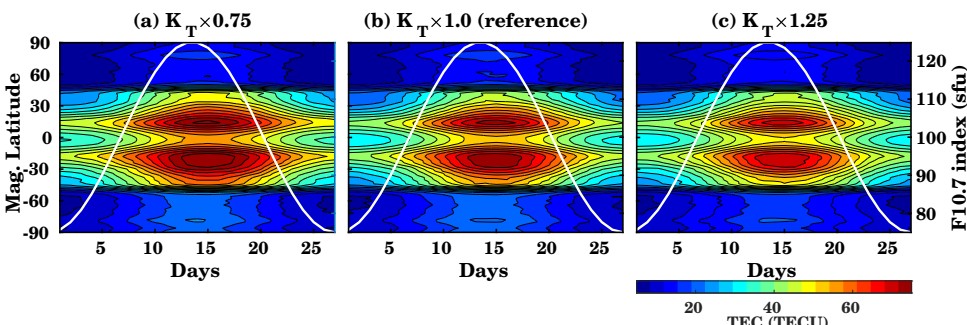

**Figure 2.** CTIPe simulated daily mean zonal mean TEC for the runs (a) $K_T \times 0.75$, (b) $K_T \times 1.0$ (reference), and (c) $K_T \times 1.25$. The white curves show the input F10.7 index.

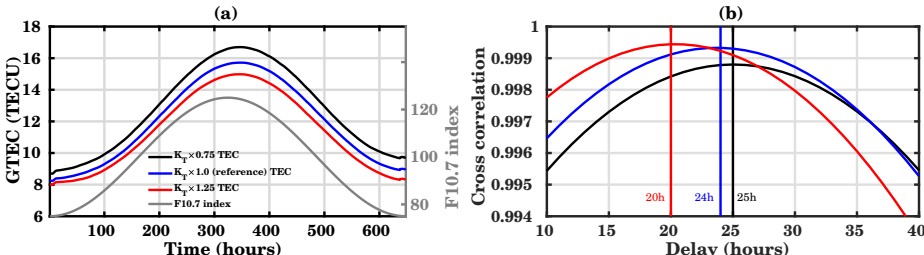

**Figure 3.** (a) Time series of modeled GTEC for different runs (a) $K_T \times 0.75$, (b) $K_T \times 1.0$ (reference), and (c) $K_T \times 1.25$, together with F10.7 given as a gray line. (b) Cross-correlation, and the delay between global mean TEC and F10.7 for the different runs.

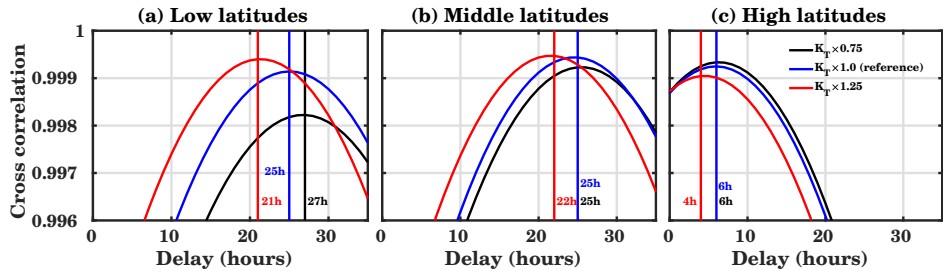

**Figure 4.** Cross-correlation and delay between regional mean TEC and the F10.7 index at low (a), middle (b), and high (c) geomagnetic latitudes for different runs.

below. The ionospheric time delay is mainly due to the imbalance between the production and loss of the ions and electrons (Ren et al., 2018).

We also analysed the model results separately for the Northern Hemisphere (NH) and the Southern Hemisphere (SH), but the differences between the hemispheres are small, and amount to 3 h, 4 h, and 4 h for the $K_T \times 0.75$, $K_T \times 1.0$, and $K_T \times 1.25$
runs, respectively (not shown).

Figure 4 shows the variation of the time delay at low [$\pm 30°$], middle [$\pm(30° - 60°)$] and high [$\pm(60° - 90°)$] geomagnetic latitudes for different eddy diffusion conditions. At low latitudes (Figure 4(a)), the delay is more sensitive to eddy diffusion than at middle and high latitudes, as this region is not only controlled by the EUV. Here, dynamics plays an essential role, especially in the equatorial ionization anomaly. Thus, small changes in eddy diffusion can lead to a more significant change
in the ionospheric delay. In general, the delay at low latitudes is longer than for the global average in Figure 3. For the run $K_T \times 1.25$, the delay is reduced by 4 h compared to the reference run.

At mid-latitudes (Figure 4(b)) the delay in the run $K_T \times 1.25$ is about 22 h, i.e. it is longer than on a global average. This is also true for the other runs where the delay is qualitatively the same and amounts to about 25 h. In this region for run $K_T \times 0.75$, the delay is similar to the one of the reference run and is about 25 h.
At high latitudes (Figure 4(c)), the variation in the delay is qualitatively the same as at middle and low latitudes, i.e., a decrease in diffusion increases the delay and vice versa. However, at high latitudes, a change in diffusion has a smaller effect,

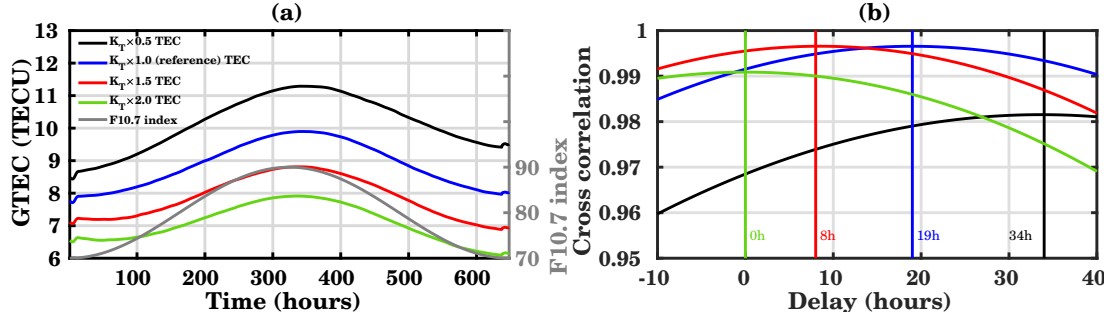

**Figure 5.** (a) Time series of modeled TEC for the different diffusion conditions $K_T \times 0.5$, $K_T \times 1.0$ (reference), $K_T \times 1.5$ and $K_T \times 2.0$. F10.7 is added as a gray line. (b) Cross-correlation, and the delay between global mean TEC and F10.7 for the different diffusion conditions.

and the delay varies between 4 h and 6 h for the different runs. For all runs, the delay is much smaller at high latitudes than at mid-latitudes. In comparison to low- and mid-latitudes, the high latitudes show less time delay in run $K_T \times 0.75$. The delay in high latitudes is also less sensitive to diffusion changes compared to the low and middle latitude regions. Similar to the
runs presented in Figure 3, the model has been run for low solar activity conditions with F10.7 in the range 70-90 sfu and using four different diffusion conditions $K_T \times 0.5$, $K_T \times 1.0$ (reference), $K_T \times 1.5$, and $K_T \times 2.0$, which amounts to 50%, 100%, 125%, and 150% of the original values in the model, respectively, as shown in Figure 5. Figure 5(a) shows the time series of TEC for different runs and the input F10.7. In comparison to Figure 3(a), the TEC values are smaller, following the F10.7 index. For these runs, the magnitude of eddy diffusion has been changed by 50%. Therefore, significant differences in
TEC size are observed. In the reference run, TEC varies from about 8 TEC to 11.3 TECU, while it shows a similar pattern for decreased/increased eddy diffusion with the difference in relative amplitude of TEC. The difference in the TEC curves in Figure 5(a) depends on the solar flux and the magnitude of the eddy diffusion coefficient. Also, the delay is calculated using the hourly TEC datasets and the F10.7 index, as shown in Figure 3(a). For the reference run $K_T \times 1.0$, the delay in the simulated GTEC is about 19 h, while the delay increases to 34 h for the run $K_T \times 0.5$, and it decreases with the increased diffusion
conditions. Here, the delay is more sensitive to the eddy diffusion compared to the 25% change cases, since the solar activity is less dominant. Compared to low solar activity, the eddy diffusion is less dominant in moderate solar activity, and the delay fluctuations are smaller. It should be noted that increasing solar activity leads to an increase in ionospheric delay.

To shed more light on the spatial patterns of the correlation between the F10.7 index and TEC, as well as on the ionospheric delay, the latter is shown in Figure 6 for each model grid point. Figure 6(b) shows the spatial map for the reference run $K_T \times 1.0$.
Maximum longitudinal differences are observed in the low and middle latitude region. Near the equatorial region, the delay varies from 10 to 40 h. At high latitudes, the delay is about 0 to 10 h.

The longitudinal variation of the delay follows the magnetic field. The maximum delay is, in line with the results in Figure 4, generally observed at lower and middle latitudes.

As is the case with GTEC, at all latitudes, the delay in local TEC is generally increased in run $K_T \times 0.75$ and decreased
in run $K_T \times 1.25$ with respect to the run $K_T \times 1.0$. In the CTIPe model, the low and mid-latitude ionosphere model and the

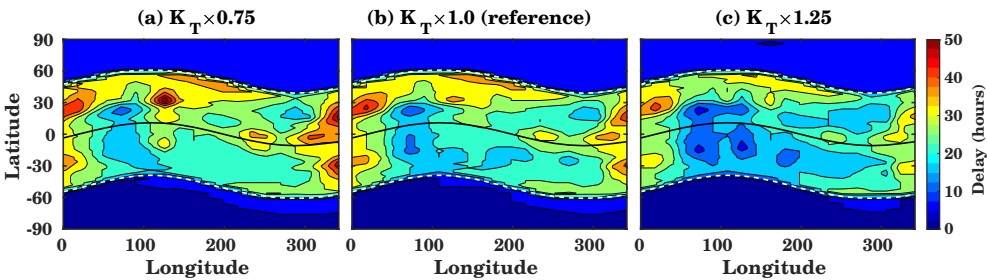

**Figure 6.** Spatial distribution of time delay between the CTIPe-TEC and the F10.7 index for different transport conditions (a) $K_T \times 0.75$, (b) $K_T \times 1.0$ (reference), and (c) $K_T \times 1.25$.The black line represents the magnetic equator, and dashed white lines represent magnetic latitudes $55°$ N/S.

high latitude ionosphere model are implemented separately. Therefore, the significant change in delay seen at $55°$ N/S may be owing to model peculiarities in CTIPe.

In the following, we investigate the height variation of the delay using the atomic oxygen ion density at geographic coordinates $40°N/18°E$ (magnetic latitude $39.06°N$). Figure 7 shows the delay between the atomic oxygen ion density and the
F10.7 index at different pressure levels. At pressure level 12 (260 km), the delay is about 24 h, 18 h, and 6 h for the different eddy diffusion cases $K_T \times 0.75$, $K_T \times 1.0$, and $K_T \times 1.25$, respectively. The delay continues to increase above pressure level 12, where it is quite close to the delay observed in TEC. This is owing to the fact that the delay observed in TEC is mainly determined by the delay of the F region, i.e., at higher pressure levels ( 200-260 km).

The eddy diffusion can influence the general circulation, and hence the thermospheric neutral species. However, the thermo-
spheric circulation is controlled not only by eddy diffusion, but also by temperature, pressure, neutral species, etc. All these parameters are affected by solar EUV radiation. To investigate how eddy diffusion affects the T/I system, we further analyze the evolution of various parameters such as the atomic oxygen number density ($n_O$), molecular oxygen density ($n_{O_2}$), molecular nitrogen density ($n_{N_2}$), molecular oxygen dissociation rates ($j_{O_2}$), neutral temperature (T), and electron density ($n_e$). Figures 8 and 9 show the variations of these parameters for the 27 day cycle for the reference run (blue color in the upper panel) and
percentage differences (lower panel: on the second y-axis) from the reference run for the other diffusion conditions $K_T \times 0.75$ and $K_T \times 1.25$, respectively, at geographic coordinates $40°N/18°E$.

Figure 8 shows the variation of $n_O$, $n_{N_2}$ and $n_{O_2}$ at pressure level 12 ($\sim 260\ km$). Eddy diffusion has a strong influence on $O$, $N_2$, and $O_2$. Figure 8(a) shows the variations of the atomic oxygen density at pressure level 12 for a 27 day solar rotation period. It shows that the atomic oxygen density decreases with increasing solar flux, connected with an increase in temperature,
which is shown in Figure 9(a). In comparison to the reference run, the percentage difference increases to about 1% for the run $K_T \times 0.75$ during the 27 day run. This is partly, but not completely due to the temperature decrease by $\sim 0.7\%$ (Figure 9(a)). Thus, reduced transport leads to reduced atomic oxygen. For the run $K_T \times 1.25$, the atomic oxygen density decreases by about 1.5%. These differences are not connected with the solar cycle, but evolve gradually over the full time interval.

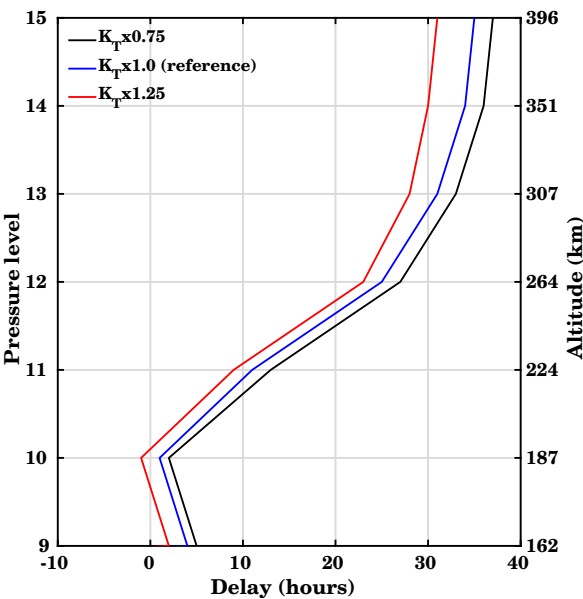

**Figure 7.** Vertical distribution of time delay between the atomic oxygen ion density and the F10.7 index for different transport conditions $K_T \times 0.75$, $K_T \times 1.0$ (reference), and $K_T \times 1.25$ at geographic coordinates $40°N/18°E$.

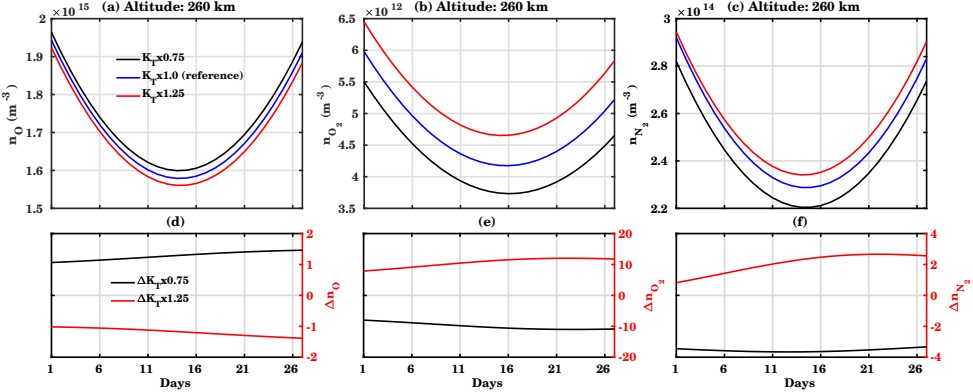

**Figure 8.** Variation of the CTIPe simulated (a) $n_O$, (b) $n_{O_2}$, and (c) $n_{N_2}$ for different diffusion conditions $K_T \times 0.75$ (black), $K_T \times 1.0$ (blue) and $K_T \times 1.25$ (red) (upper panel) for the pressure level 12 (260 km). The percentage difference between the run $K_T \times 1.0$ (blue curve) and those with modified eddy diffusion conditions $K_T \times 0.75$ (black), and $K_T \times 1.25$ (red) are shown in the lower parts of the panels.

Similar to the atomic oxygen density variations, the molecular oxygen and nitrogen densities also decrease with increasing solar flux (Figure 8(b) and 8(c)). For the molecular oxygen density, the percentage difference decreases to about 10% for the run $K_T \times 0.75$, while it increases to about 10% for the $K_T \times 1.25$ run. Similar variations are observed in the behaviour of the molecular nitrogen density (Figure 8(c)). Once the diffusion increased, the $n_{O_2}$ increases compared to the reference run, demonstrating that diffusion is a critical process to control the evolution of oxygen. Therefore, we register a change in the total composition due to an increase or decrease in eddy diffusion.

Figure 8(d) and 8(b) shows the percentage difference between the reference run results and those of the runs with increased or decreased eddy diffusion. For the run $K_T \times 0.75$, the atomic oxygen density increases to about 1%, while the molecular oxygen decreases by 10%. Similar to the molecular oxygen, the molecular nitrogen density also decreases by $\sim 3\%$. In comparison to $K_T \times 0.75$, opposite trends can be seen for the run $K_T \times 1.25$.

In Figure 9(a) and 9(b), the 27 days behaviour at an altitude of $\sim 260\ km$ is shown for T and $n_e$. T increases with increasing solar irradiance. As an increase in solar irradiance expands the range of the thermosphere region, the scale height of each component changes. An increase in solar radiation flux will also increase the height of each pressure level. In Figure 9(e), non-monotonic variations are observed in the difference between the reference run and $K_T \times 1.25$. This could be due to the combined effect of different diffusion cases and solar flux. Compared to the reference case, the temperature decreases by about 0.7% for the $K_T \times 0.75$ run, while it increases by 0.7% for the $K_T \times 1.25$ run. Similar to $T$, $n_e$ also varies with the solar flux. An increase in the solar radiation flux leads to an increase in ionization and thus to an increase in electron density.

The $j_{O_2}$ also vary for different diffusion conditions, as shown in Figure 9(c) for pressure level 7 (altitude $\sim 125 km$). An increase in eddy diffusion reduces $j_{O_2}$, leading to an increase in $n_{O_2}$ and a reduction in $n_O$. Exactly the opposite behaviour is observed for a decrease in eddy diffusion.

Since we are dealing with vertical transport processes, it is essential to analyze the latitudinal variation against pressure levels. Figure 10 shows the percentage difference of $T$, $j_{O_2}$ and $n_e$ in the runs $K_T \times 0.75$ and $K_T \times 1.25$ with respect to $K_T \times 1.0$ for the $14^{th}$ model day. Figure 10(b) and 10(c) show that due to a decrease/increase in eddy diffusion $T$ decreases/increases at all pressure levels.

The lowest four pressure levels belong to the lower boundary, where the neutral wind, temperature, and height of the pressure level are imposed as boundary conditions from the WAM model. An increase of the eddy diffusion by a factor of 25% ($K_T \times$ 1.25) leads to an increase in $T$ by 1%. It mainly affects pressure levels 7-9 (125-160km). The percentage difference in $T$ is negligible at pressure levels 5-6 ( 110 km ), but the variations increase with altitude. Figure 10(d), shows the latitude-pressure distribution of $n_e$. For the run $K_T \times 0.75$, it shows that for a reduction in eddy diffusion, $n_e$ is increased in the thermosphere above pressure level 9 (160 km). Interestingly, above this altitude,$n_e$ increases by about 7 %. Electron density increases in the low-latitude region at pressure level 4 (98 km) and in the high-latitude region at pressure level 5 (105 km). The response of the thermosphere $n_e$ to an enhancement of eddy diffusion is entirely different. For the run $K_T \times 1.25$, $n_e$ decreases at higher pressure levels but it increases at lower pressure levels, except for mid-latitudes at 98 km and high latitudes at 105 km.

The variation in $j_{O_2}$ is shown in Figure 10(g). The percentage difference for the run $K_T \times 0.75$ compared to the reference run decreases by about 7% for pressure levels 5-7 (105-125 km), and it decreases by 7% for the run $K_T \times 1.25$.

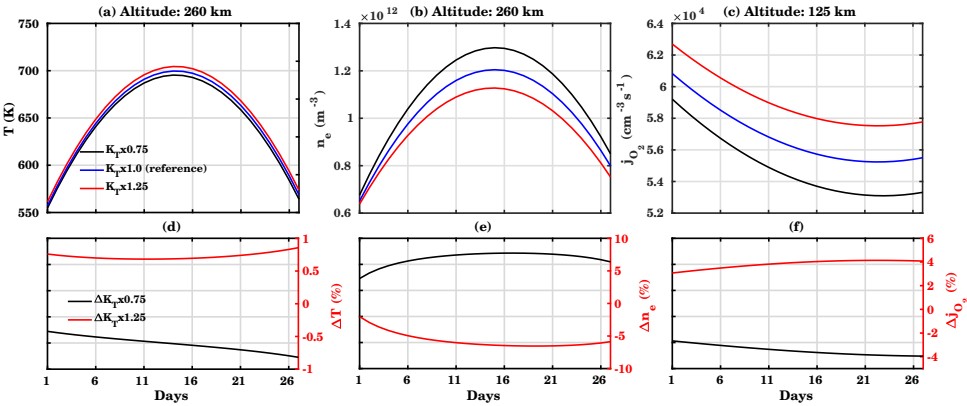

**Figure 9.** Same as Figure 8 for (a) T, (b) $n_e$ for the pressure level 12 ($\sim 260\ km$), and (c) $j_{O_2}$ for pressure level 7 ($\sim 125\ km$).

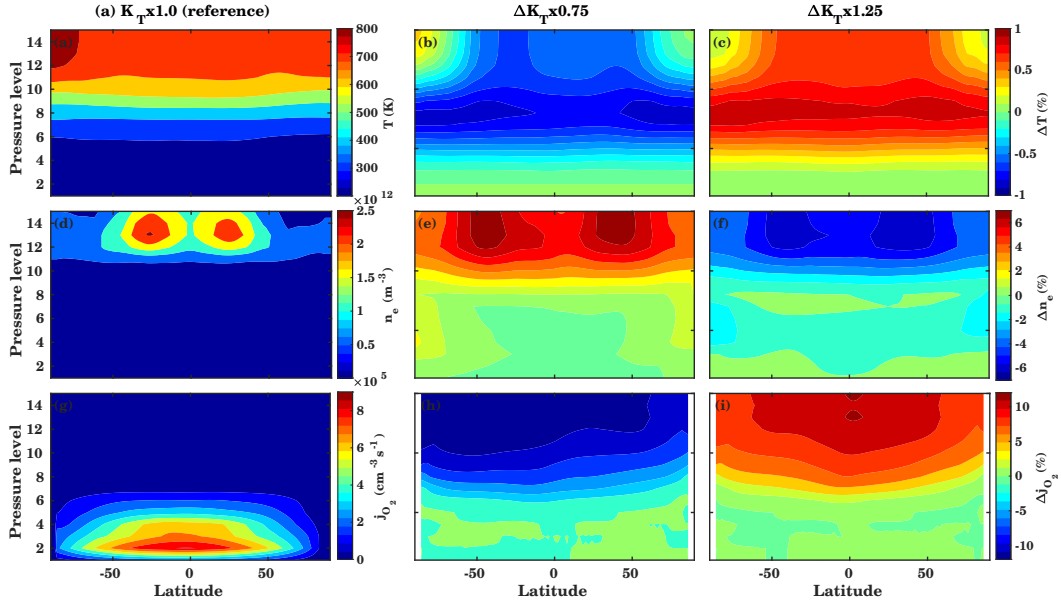

**Figure 10.** (a) CTIPe simulated $T$ (upper row), $n_e$ (middle row), and $j_{O_2}$ (bottom row), for the $14^{th}$ model day. The other columns show the percentage difference between the $K_T \times 1.0$ simulation and those with modified eddy diffusion conditions: (b) $K_T \times 0.75$ and (c) $K_T \times 1.25$.

Figure 11(a) shows the variation of $n_O$. For the run $K_T \times 0.75$, $n_O$ is increased by 5-7% above the turbopause. The enhanced diffusion leads to an increase of $n_O$ in the lower thermosphere due to the downward transport of $n_O$ from higher altitudes (Rees and Fuller-Rowell, 1988). Note that eddy diffusion has a more substantial impact at high latitudes below the turbopause. Chandra and Sinha (1974) showed that due to photochemical effects, the variation of eddy diffusion does not contribute significantly to $n_O$ below 100 km, but above 100 km it decreases with increasing eddy diffusion.

Enhanced eddy diffusion leads to an increase in $n_{O_2}$ of about 10-12% above the turbopause in the $K_T \times 1.25$ run, with $j_{O_2}$ decreasing by about 0.5%, as shown in Figure 10(d). Thus, the decrease in $j_{O_2}$ increases $n_{O_2}$, and this leads to a decrease in $n_O$. Similar variations are also observed in the case of enhanced diffusion conditions for $n_{N_2}$, with an increase of about 2-3% for the enhanced eddy diffusion conditions. The variation in eddy diffusion affects the composition at different altitudes through molecular diffusion.

Enhanced eddy diffusion leads to an increase in molecular oxygen. This leads to a decrease in atomic oxygen at all altitudes above 100 km due to molecular diffusion. As a result, there is a significant decrease in the $[O]/[N_2]$ ratio. Qian et al. (2009) studied the effect of modified eddy diffusion on thermospheric composition using the NCAR TIE-GCM model and reported similar results. These simulations revealed a new finding how eddy diffusion can strongly affect the thermospheric composition ($O$, $O_2$, and $N_2$) through the ionospheric delay.

The steady-state electron density N can be written according to Rishbeth (1998):

$$N \sim \frac{q}{\beta} \sim \frac{I[O]}{\gamma_1[N_2] + \gamma_2[O_2]} \tag{6}$$

where $q$, $\beta$, I, and $\gamma_1$, $\gamma_2$ are the production term, the loss term, the solar ionizing flux, and the reaction rates, respectively.

The composition of the T/I system is mainly controlled by various production and loss mechanisms. The production of electrons is mainly due to the ionization of atomic oxygen through solar EUV, and the loss is mainly controlled by $N_2$. The production of atomic oxygen ions depends not only on the atomic oxygen density but also on the solar radiation. Ren et al. (2018) explained that the delay observed in the electron density depends on the production and loss processes as well as the $[O]/[N_2]$ ratio. The major loss of ions in the F regions is given by the following reactions:

$O^+ + N_2 \longrightarrow NO^+ + N$

$O^+ + O_2 \longrightarrow O_2^+ + O$

The rate coefficients $\gamma_1$ and $\gamma_2$ in Eq.(6) are given, e.g., by St.-Maurice and Torr (1978). These reaction rate coefficients are dependent on the effective temperature ($T_f$), which significantly affects the loss reaction and composition:

$$T_f = 0.63 \times T_i + 0.36 \times T_N. \tag{7}$$

Here $T_i$ and $T_N$ are ion temperature and the neutral temperature, respectively. For low values of $T_f < 1100K$, the loss rate coefficients $\gamma_1$ and $\gamma_2$ decrease with increasing $T_f$, while for $T_f > 1100K$, the loss rate $\gamma_1$ increases as a result of the electron density decrease with increasing F10.7 index. The nonlinear relation between the loss rate coefficients and $T_f$ is shown by Su et al. (1999).

Figure 12 shows the variations of the $[O]/[N_2]$ ratio for different diffusion conditions at geographical lat/lon $40°N/18°E$ at an altitude of about 260 km (pressure level 12). For the reference run, the delay is about 2-3 days, since the peak response is observed at day 16. The $[O]/[N_2]$ ratio strongly decreases with increasing eddy diffusion, and the delay is also shifted to one day. Thus, the variation in eddy diffusion strongly affects the $[O]/[N_2]$ ratio, which in turn affects the delay mechanism.

Figure 13 shows the effect of eddy diffusion on the atomic oxygen ionization (a) and loss rates (c) through molecular nitrogen at $40°N/18°E$ and the difference between the reference run and other diffusion cases are shows in Figure 13(b) and 13(d). The reference case $K_T \times 1.0$ and the runs $K_T \times 0.75$ and $K_T \times 1.25$ are represented by blue, black, and red curves, respectively. The maximum ionization occurs at pressure level 9-10 (162-187 km) (Figure 13(a)). Figure 13(b) shows a decrease of ionization rates with enhanced eddy diffusion, whereas they are increased for reduced eddy diffusion. The production term in Eq.(6) depends strongly on the ionization rates and the atomic oxygen density. Therefore, increased eddy diffusion decreases ionization and atomic oxygen density. Figure 13(d) shows that the loss rates are reduced by about 0.5 % in the F region in the case of enhanced eddy diffusion. Su et al. (1999) discussed the dependence of the loss rates on temperature. They suggested that the loss rate coefficient decreases with increasing $T_f$. Enhanced eddy diffusion leads to an increase in molecular components while reducing atomic oxygen.

Consequently, enhanced $N_2$ increases the overall loss term in Eq.(6) and reduces the electron density, resulting in a reduced delay in TEC. Based on the model simulations, we conclude that eddy diffusion is one of the major factors responsible for the changes in thermospheric composition via general circulation and significantly affects the ionospheric delay. Although the current investigation suggests that a small change in loss rates can affect the delay for several hours, further numerical modeling using real observations and varying atmospheric conditions is needed to understand the physical processes.

## 4   Summary

Using a 1-D model, Jakowski et al. (1991) first reported that the delayed density variation concerning solar EUV variations is probably due to the slow diffusion of atomic oxygen. Based on their hypothesis, the ionospheric delay in TEC, simulated by the CTIPe model, was investigated. Using the F10.7 index, the ionospheric delay at the solar rotation period is well reproduced and amounts to about 1 d (Jacobi et al., 2016; Schmölter et al., 2018). The thermosphere/ionosphere coupling plays an important role in the delay mechanism and this was reported in several studies, but it was barely investigated. Therefore, this is the first time we investigated the impact of eddy diffusion on the ionospheric delay. To investigate the physical mechanism of ionospheric delay at the solar rotation period, we performed various experiments using CTIPe model. From the mechanistic studies using CTIPe, results show that eddy diffusion is an important factor that strongly influences the delay introduced in TEC based on the solar activity conditions. In the case of reduced eddy diffusion to 75% of the original value, the delay is slightly longer (about 25 h), while in the case of increased transport the delay is reduced to 20 h. An increase in eddy diffusion leads to faster transport processes and an increased loss rate, resulting in a reduction of the ionospheric time delay.

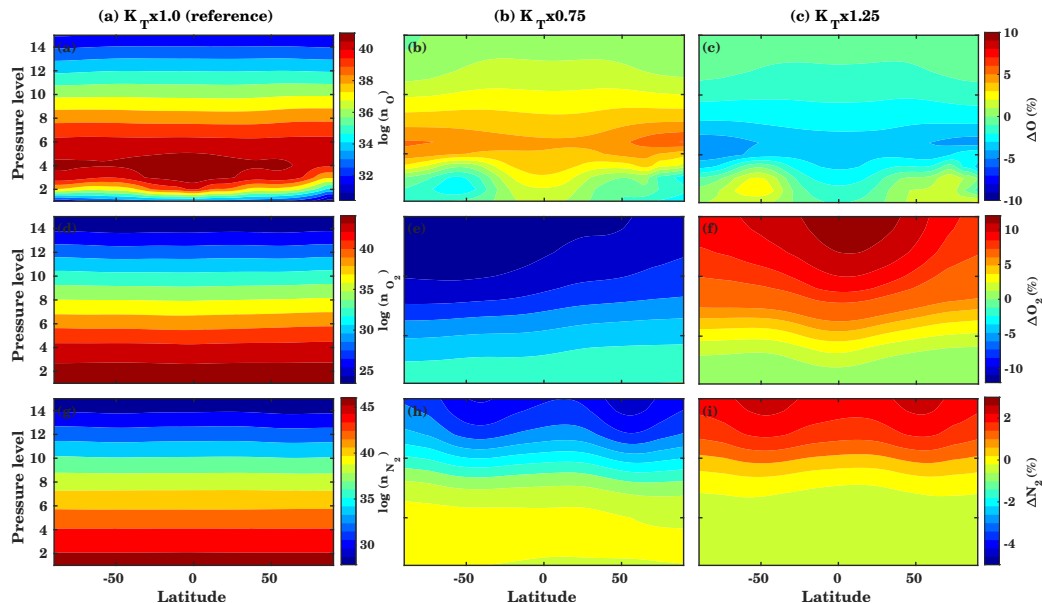

**Figure 11.** Same as Figure 10, but, for $n_O$, $n_{O_2}$, and $n_{N_2}$.

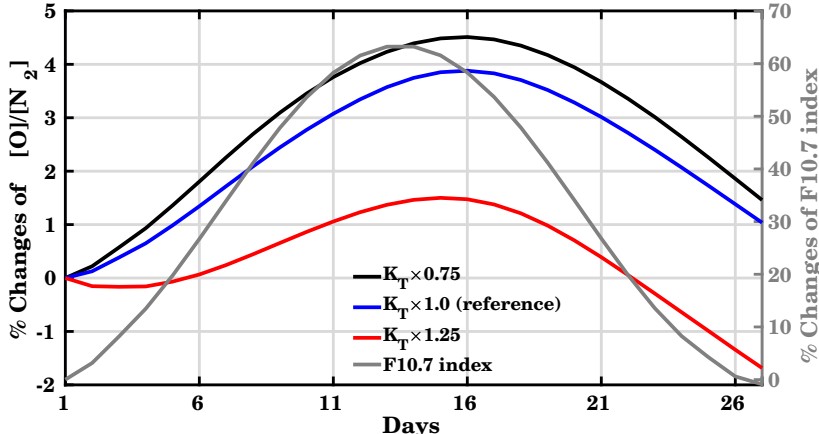

**Figure 12.** Percentage change of $[O]/[N_2]$ ratio for different diffusion conditions $K_T \times 0.75$, $K_T \times 1.0$ (reference), and $K_T \times 1.25$ at pressure level 12 at geographic coordinates $40°N/18°E$. The right y-axis refers to the percentage change of the F10.7 index (gray curve).

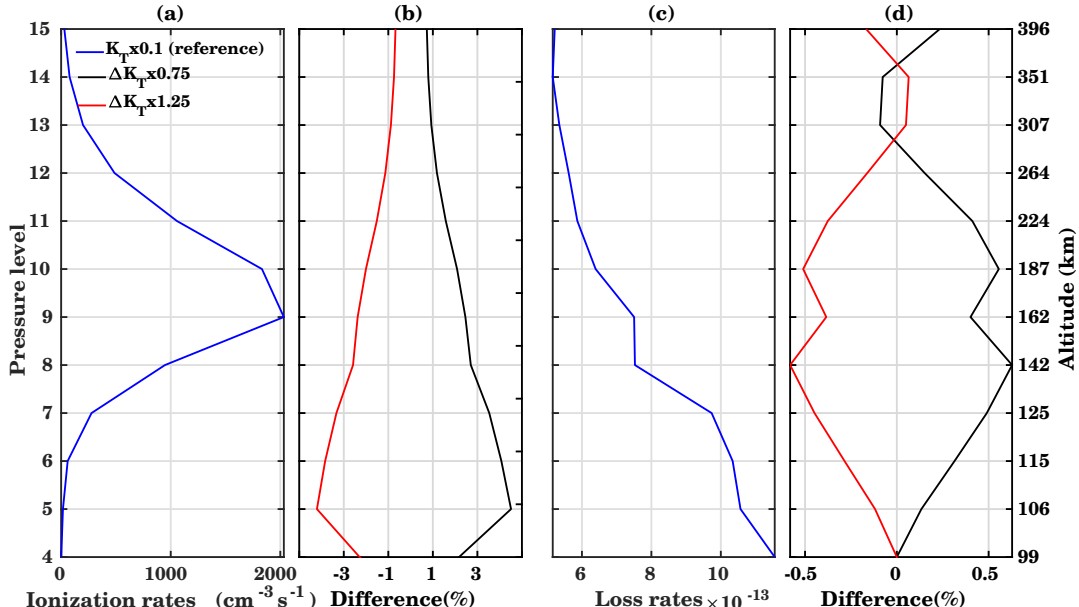

**Figure 13.** Atomic oxygen ionization (a) and loss rates (c) due to molecular nitrogen for the reference run $K_T \times 1.0$ and its difference (b,d) with $K_T \times 0.75$ and $K_T \times 1.25$ at different pressure levels on the $14^{th}$ model day at geographic coordinates $40°N/18°E$.

At low latitudes, the influence of solar activity is stronger, as EUV radiation drives ionization processes that lead to compositional changes. Therefore, the combined effect of eddy diffusion and solar activity shows more delay in the low and mid-latitude region.

Our results suggest that eddy diffusion plays a crucial role in the ionospheric delay. Therefore, further numerical modeling and observational results are required to better understand the role of lower atmospheric forcings and thermosphere/ionosphere coupling.

For this study, constant atmospheric conditions have been used to understand the role of solar flux and eddy diffusion in the ionospheric delay. In future, further investigation is required to explore the physical processes using actual observations. It would also be interesting to investigate the combined effect of solar variations, geomagnetic variations, and lower atmospheric forcings.

*Author contributions.* RV together with CJ and MC performed the CTIPe model simulations. RV drafted the first version of the manuscript. All authors discussed the results and contributed to the final version of the manuscript.

*Competing interests.* Christoph Jacobi is one of the Editors-in-Chief of Annales Geophysicae. The authors declare that they have no conflict of interest.

*Acknowledgements.* The study has been supported by Deutsche Forschungsgemeinschaft (DFG) through grants nos. BE 5789/2-1 and JA 836/33-1.

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
