# Peer review of "Role of Eddy Diffusion in the Delayed Ionospheric Response to Solar Flux Changes"

_Annales Geophysicae, 2021_

## Author Comment (AC1)

**Reply to First Reviewer's Comments:**

This study focused on the contributions from the eddy diffusion to the ionospheric delay, which is an interesting topic and worthy of investigation. The authors showed that the increased eddy diffusion can contribute to decreasing the delay of the ionosphere. The results are not surprising but this work helps to improve our understanding of the ionospheric delayed response to the periodic variation of solar activity to some degree. There are some issues which should be addressed before the possible publication

Response: We are thankful for the reviewer's comments and suggestions which help us to improve the quality of the manuscript. We will address all the raised points in the revised version of the manuscript.

The authors mentioned "…the eddy diffusion is a primary factor that influences the delay of the ionospheric total electron content (TEC)" in many places. It should be noted that the root mechanism that responsible for the ionospheric delay is the imbalance between the production and loss of the electrons. The eddy diffusion can alter the general circulation, which thus impact the thermospheric neutral species. Whereas, the thermospheric circulation is controlled by not only eddy diffusion, but also temperature, pressure, neutral species, et al. All of them are affected by solar flux changes. Thus, it is not true to say that the eddy diffusion is the "primary factor". The author should clarify this statement throughout the paper.

Response: Thank you. We agree with the reviewer's suggestion. We will improve this in the revised text and replace "a primary" by "an important", and we will make clear in the text that production/loss imbalance is the primary mechanism.

Different eddy diffusion coefficients can result in different ionospheric time delay. However, the eddy diffusion at a given location and time interval should NOT change much. The eddy diffusion could contribute to the long-term change of ionospheric time delay, for instance, the seasonal variation of time delay.

Response: Thank you for this comment. In this paper, our effort was to understand the relation between the eddy diffusion coefficient and ionospheric time delay using model simulations. We agree with the reviewer's concern in general. But several authors have reported that the eddy diffusion may change during months or seasons, but these observations are available for specific locations and time period (e.g., Sasi and Vijayan, 2001). We choose to perform the model simulations for the solar rotation period and found direct impact on the ionospheric delay.

Reference: Sasi, M. N. and Vijayan, L.: Turbulence characteristics in the tropical mesosphere as obtained by MST radar at Gadanki (13.5°N, 79.2°E), Annales Geophysicae, 19, 1019–1025, https://doi.org/10.5194/angeo-19-1019-2001, 2001.

It seems that the authors did not show the results for the first day and 27th day in Figures 1 and 8-9. I suggest them to plot the results during the whole 27-day interval. In addition, it should be better to draw the vertical line at 13.5 days instead of the 14th day.

Response: Thank you. We have added the results for the first day and 27th day, and we will improve the figure for clarity and include a vertical line at 13.5 day.

There are two relevant papers which can be cited in this work. (1) Schmölter, E., Berdermann, J., & Codrescu, M. (2021). The delayed ionospheric response to the 27â day solar rotation period analyzed with GOLD and IGS TEC data. *Journal of Geophysical Research: Space Physics*, 126, e2020JA028861. https://doi.org/10.1029/2020JA02886 (2) Ren, D., Lei, J., Wang, W., Burns, A., & Luan, X. (2021). Observations and simulations of the peak response time of thermospheric mass density to the 27â day solar EUV flux variation. *Journal of Geophysical Research: Space Physics*, 126, e2020JA028756. https://doi.org/10.1029/2020JA028756

Response: Thank you for the suggestions. We will include these references in the revised version.

---

## Author Response (AR1)

**Reply to First Reviewer's Comments:**

This study focused on the contributions from the eddy diffusion to the ionospheric delay, which is an interesting topic and worthy of investigation. The authors showed that the increased eddy diffusion can contribute to decreasing the delay of the ionosphere. The results are not surprising but this work helps to improve our understanding of the ionospheric delayed response to the periodic variation of solar activity to some degree. There are some issues which should be addressed before the possible publication

Response: We are thankful for the reviewer's comments and suggestions which help us to improve the quality of the manuscript. We have revised the paper according to the suggestions and comments.

We discussed most of the comments of reviewer#1 in the first response. Here we summarise the important points which we have been included in the revised version.

The authors mentioned "...the eddy diffusion is a primary factor that influences the delay of the ionospheric total electron content (TEC)" in many places. It should be noted that the root mechanism that responsible for the ionospheric delay is the imbalance between the production and loss of the electrons. The eddy diffusion can alter the general circulation, which thus impact the thermospheric neutral species. Whereas, the thermospheric circulation is controlled by not only eddy diffusion, but also temperature, pressure, neutral species, et al. All of them are affected by solar flux changes. Thus, it is not true to say that the eddy diffusion is the "primary factor". The author should clarify this statement throughout the paper.

Response: Thank you for the suggestion. We have improved this in the revised text and replace "a primary" by "an important", and we have improved this throughout the manuscript.

Different eddy diffusion coefficients can result in different ionospheric time delay. However, the eddy diffusion at a given location and time interval should NOT change much. The eddy diffusion could contribute to the long-term change of ionospheric time delay, for instance, the seasonal variation of time delay.

Response: Thank you for this comment. In this paper, our effort was to analyze the relation between the eddy diffusion coefficient and ionospheric time delay using model simulations. We agree with the reviewer's concern in general. But several authors have reported that the eddy diffusion may change during months or seasons, and their observations are available for specific locations and time period (e.g., Sasi and Vijayan, 2001). We choose to perform the model simulations for the solar rotation period and found direct impact on the ionospheric delay.

Reference: Sasi, M. N. and Vijayan, L.: Turbulence characteristics in the tropical mesosphere as obtained by MST radar at Gadanki (13.5°N, 79.2°E), Annales Geophysicae, 19, 1019–1025, https://doi.org/10.5194/angeo-19-1019-2001, 2001.Page: 5, Lines: 132-135

It seems that the authors did not show the results for the first day and 27th day in Figures 1 and 8-9. I suggest them to plot the results during the whole 27-day interval. In addition, it should be better to draw the vertical line at 13.5 days instead of the 14th day.

Response: Thank you. We have added the results for the first day and 27th day, and we have improved the figure for clarity and include a vertical line at 13.5 days. Pages: 6, 11 and 13.

There are two relevant papers which can be cited in this work. (1) Schmölter, E., Berdermann, J., & Codrescu, M. (2021). The delayed ionospheric response to the 27â day solar rotation period analyzed with GOLD and IGS TEC data. *Journal of Geophysical Research: Space Physics*, 126, e2020JA028861. https://doi.org/10.1029/2020JA02886 (2) Ren, D., Lei, J., Wang, W., Burns, A., & Luan, X. (2021). Observations and simulations of the peak response time of thermospheric mass density to the 27â day solar EUV flux variation. *Journal of Geophysical Research: Space Physics*, 126, e2020JA028756. https://doi.org/10.1029/2020JA028756

Response: Thank you for the suggestions. We have included these references in the revised version. Page: 3 Lines: 87-89, 71-72

**Reply to Second Reviewer's Comments:**

This manuscript successfully explores the role of eddy diffusion in controling the degree of ionospheric response delay to solar activity driving by undertaking a simple, but reasonably rigorous, set of simulations with CTIPe. The results fit well within the context of existing work and I belive these results to provide new insight into the behaviour of this ionospheric time delay.

It is quite rare for me to do this (perhaps my first time doing this), but I see no significant issues with this manuscript. While this may seem like somewhat of a cop-out of a review, the previous reviewer has already covered any concerns I might have had for this manuscript and I believe that the manuscript is now sufficiently clear and conclusive enough to warrant publication after they complete the modifications requested by the previous reviewer.

Response: We are thankful to the reviewer for reviewing our manuscript and providing a very positive opinion. We have addressed all the raised points suggested by the referees in the revised version of the manuscript.